Radar remote sensing-based inversion model of soil salt content at different depths under vegetation

Chen Yinwen 1
Du Yuyan 2 1454902408@qq.com
Yin Haoyuan 3 4
Wang Huiyun 3 4
Chen Haiying 1
Li Xianwen 3 4
Zhang Zhitao 3 4
Chen Junying 3 4
1 College of Language and Culture, Northwest A&F University , Yangling, Shaanxi , China
2 Gansu Water Conservancy & Hydro Power Survey & Design Research Institute , Lanzhou, Gansu , China
3 College of Water Resources and Architectural Engineering, Northwest A&F University , Yangling, Shaanxi , China
4 Key Laboratory of Agricultural Soil and Water Engineering in Arid and Semiarid Areas, Ministry of Education, Northwest A&F University , Yangling, Shaanxi , China
Moffat Ian
Electronic publication date: 2022 Apr 26
Publication date: 2022
Volume: 10
Electronic Location ID: e13306
Received 2021 Oct 13; Accepted 2022 Mar 30
Copyright: © 2022 Chen et al.
Copyright year: 2022
Copyright holder: Chen et al.
License: This is an open access article distributed under the terms of the Creative Commons Attribution License, which permits unrestricted use, distribution, reproduction and adaptation in any medium and for any purpose provided that it is properly attributed. For attribution, the original author(s), title, publication source (PeerJ) and either DOI or URL of the article must be cited.
License URL: https://creativecommons.org/licenses/by/4.0/

Keywords: vegetation coverage, Soil salt content, Radar remote sensing, Soil at different depths, Best subset selection, Support vector machine

Funding: POA-based Teaching Design and Practice of Comprehensive English JY2103208 Translation Program of Materials on Agricultural Science and Culture G202008-02 National Key Research and Development Program of China 2017YFC0403302 National Natural Science Foundation of China 51979232 Natural Science Basic Research Program of Shaanxi Province 2019JM066 This work was supported by POA-based Teaching Design and Practice of Comprehensive English (JY2103208), the Translation Program of Materials on Agricultural Science and Culture (No. G202008-02), the National Key Research and Development Program of China (No. 2017YFC0403302), National Natural Science Foundation of China (No. 51979232), and the Natural Science Basic Research Program of Shaanxi Province (2019JM066). The funders had no role in study design, data collection and analysis, decision to publish, or preparation of the manuscript.

==============================
Excessive soil salt content (SSC) seriously affects the crop growth and economic benefits in the agricultural production area. Prior research mainly focused on estimating the salinity in the top bare soil rather than in deep soil that is vital to crop growth. For this end, an experiment was carried out in the Hetao Irrigation District, Inner Mongolia, China. In the experiment, the SSC at different depths under vegetation was measured, and the Sentinel-1 radar images were obtained synchronously. The radar backscattering coefficients (VV and VH) were combined to construct multiple indices, whose sensitivity was then analyzed using the best subset selection (BSS). Meanwhile, four most commonly used algorithms, partial least squares regression (PLSR), quantile regression (QR), support vector machine (SVM), and extreme learning machine (ELM), were utilized to construct estimation models of salinity at the depths of 0–10, 10–20, 0–20, 20–40, 0–40, 40–60 and 0–60 cm before and after BSS, respectively. The results showed: (a) radar remote sensing can be used to estimate the salinity in the root zone of vegetation (0-30 cm); (b) after BSS, the correlation coefficients and estimation accuracy of the four monitoring models were all improved significantly; (c) the estimation accuracy of the four regression models was: SVM > QR > ELM > PLSR; and (d) among the seven sampling depths, 10–20 cm was the optimal inversion depth for all the four models, followed by 20–40 and 0–40 cm. Among the four models, SVM was higher in accuracy than the other three at 10–20 cm (RP2 = 0.67, RMSEP = 0.12%). These findings can provide valuable guidance for soil salinity monitoring and agricultural production in the arid or semi-arid areas under vegetation.

Introduction

Soil salinization seriously affects agricultural production and economic efficiency of land resources (Harti et al., 2016). In recent years, soil degradation caused by soil salinization has become a global problem due to natural environmental changes and irrational human activities (Besser et al., 2017). Rapid and comprehensive access to soil salt content (SSC) is essential for local ecological environment protection and agricultural production (Wang et al., 2019a). Currently, the use of satellite remote sensing for SSC monitoring has become a popular research direction (Wu et al., 2018; Hassani, Azapagic & Shokri, 2020).

At present, the commonly used methods include classification and interpretation of remote sensing images (Gao et al., 2016), spectral index (Wang et al., 2021b; Gorji et al., 2020; Chen et al., 2020a), and feature space analysis (Guo, Zang & Zhang, 2020). Among these methods, spectral index is a convenient and efficient SSC monitoring method (Ma et al., 2020; Habibi et al., 2021). Chen et al. (2015) significantly improved the SSC inversion accuracy by adding short-wave infrared to the traditional vegetation index. El Harti et al. (2016) improved the SSC inversion accuracy in the Morocco irrigation area by adding blue band to salt index to construct a new salt index OLI-Sr. Despite the relatively satisfactory monitoring results, these studies mostly focused on the surface soil.

Compared with optical and hyperspectral remote sensing satellites, radar remote sensors enjoy all-weather capability and short follow-up observation time and become quite promising in SSC monitoring (El Harti et al., 2016). With the data of fully polarized synthetic aperture radar, Nurmemet et al. (2015b) extracted and classified the salinity at 0–10 cm in the delta oasis in the northwest Xinjiang, China. Guo (2014) compared and analyzed the relationship between radar backscattering coefficients combined from different polarizations from Radarsat-2 images and the topsoil salinity in Hetao Irrigation District (HID), and concluded that (HH2 + HV2)/(HH2 − HV2) was the best polarization combination for salinity information extraction. Liu et al. (2016) and Liu (2014) used four-polarization radar backscattering coefficients from Radarsat-2 images to establish an artificial intelligence model for the topsoil salinity inversion in HID. Nurmemet et al. (2015a) studied the monitoring effect of PALSAR data on soil salinity at 0–20 cm in the area along Keriya River in Xinjiang, China, and proposed that the support vector machines (SVM) was the optimal model of the pixel-based classification methods. Zhang et al. (2020) applied Sentinel-1 radar images to salinity inversion modelling for the bare soil at 0–10 and 10–20 cm, respectively, and achieved satisfying inversion accuracy.

Remote sensing data, prone to factors such as land surface and atmospheric conditions, is obviously time and region sensitive. Therefore, screening sensitive polarization combination indexes is significant for salinity monitoring. At present, the commonly used variable screening methods include gray correlation (Wang et al., 2018), ridge regression (Nabiollahi et al., 2021), Lasso regression (Wang et al., 2019b), and variable importance in projection (Wei et al., 2020), but these methods only involve local optimal screening. The best subset selection (BSS), however, enumerates all possible combinations in the construction of global optimal model and uses the least free variables to explain the dependent variables so as to eliminate collinearity effect. Studies have found BSS can greatly improve the model accuracy (Zhang et al., 2019; Zhang et al., 2021; Chen et al., 2020b), indicating its feasibility for spectral index screening.

In order to simplify SSC inversion, scholars have applied machine learning methods together with radar data to soil salinity inversion (Jiang et al., 2017; Nawar et al., 2014). These methods include support vector machine (SVM), extreme learning machine (ELM), partial least squares regression (PLSR), quantile regression (QR), and so forth. Their increasing popularity in modeling different soil properties such as SSC (Szabó et al., 2019) comes from their accuracy, stability and convenience as well as the good match of their estimates with the resolution of remote sensing data (Zaman, Mckee & Neale, 2012). The application of PLSR to mapping soil salinity distribution has revealed that PLSR outperforms step multiple regression (SMR) in prediction accuracy (Sidike, Zhao & Wen, 2014; Farifteha et al., 2007). This is because the significant linear relationship between independent and dependent variables enables PLSR to be better applied to SSC inversion. Compared with other learning methods, ELM can reduce the computing time of feature extraction and prediction and improve learning efficiency (Ramendra et al., 2018; Melloa et al., 2013). For example, Lao et al. (2021) constructed a prediction model of surface SSC with ELM, and the R2 of the optimal model reached 0.93. QR is widely used because it requires no random perturbation or normal variation of variables in the model and the outliers have little influence on the overall model accuracy. The comparison of artificial neural network (ANN), SVM and QR in the performance of SSC inversion based on GF-1 satellite data has shown that QR has the highest accuracy (Zhang et al., 2019). Compared with other methods, SVM can largely overcome such problems as “large discrete values” and “overlearning” (Schug et al., 2018). The experiment of Hoa et al. (2019) has demonstrated the feasibility of using SVR together with Sentinel-1 SAR data (VV, VH and their derived textures) to map SSC in semi-arid areas.

Although these studies have established satisfactory monitoring models, they have mainly focused on salinity inversion in bare soil because of the limited penetration of microwave signal. The signal will be further weakened under vegetation. Therefore, radar satellite remote sensing cannot be used directly to estimate the salinity at different depths under vegetation.

In order to explore this practical problem, we hypothesized that radar remote sensing can monitor soil salinity at different depths under vegetation in that the soil salinity at different depths may act on the vegetation that will in turn influence radar signals. We aimed to (1) relate radar backscattering coefficients to soil salinity at different depths via vegetation; (2) analyze the relationship between radar backscattering coefficient and soil salinity at different depths; and (3) compare the estimation accuracy of four machine learning models so as to construct highly accurate salinity inversion models for different soil depths.

Materials and Methods

Overview of research site

The experiment was conducted in the Shahaoqu Irrigation Area (SIA) of the Hetao Irrigation District, Inner Mongolia, China (Fig. 1). SIA totals about 6,000 hectares (15 km from north to south and 4 km from west to east). The south is higher than the north, and the average altitude is 1,034–1,037 m. The soil samples were identified as silty clay loam and sandy loam according to the USDA system (Soil Texture Calculator NRCS, https://www.nrcs.usda.gov). The main vegetation includes corn, wheat and sunflower. Located in the mid-temperate zone, SIA is slightly influenced by the warm southeast monsoon and greatly affected by the dry and cold northwest monsoon. This area is cold and dry all the year round, with an average temperature of about 7 °C, an average annual precipitation and evaporation of 140 and 2,000 mm, respectively (Zhang et al., 2018). The irrigation method here is mainly flood irrigation from the Yellow River, supplemented by submersible irrigation. Long-term inappropriate irrigation has led to secondary salinization in this area.

Figure 1 Distribution of the sampling points at the research site.

Soil sample collection and chemical analysis

This research is supported by the program of Diagnosing and Estimating Techniques and Methods of Soil Moisture and Salt Content at Different Depths, a subitem of the State Key Development Program of Water Saving, Sewage Reduction and Salinity Control Technology and its Application in the Farmland, which was approved by Chinese Ministry of Science and Technology on March 29, 2020. HID administration approved the experiment in accordance with the program No. (2017YFC0403302) rather than issuing a field permit. The soil samples were collected from July 15 through 18, 2019 when SIA was covered with vegetation. In selecting the sampling points, such local conditions as salinity and vegetation coverage were taken into consideration. The five-point sampling method was adopted, and a hand-held ring knife was utilized to conduct stratified sampling at 0–10, 10–20, 20–40 and 40–60 cm. GPS was used to record the longitude and latitude information of the sampling points. In total, 114 sampling plots (each plot has five sampling points) were established, and the soil samples were collected into the prepared aluminum boxes, weighed and carried to the laboratory. The obtained soil samples were placed in the oven at 105 °C for 8 h. to be dried completely, and then the dried samples were weighed again and ground to remove the large particles. Next, the soil solution (the mass ratio of distilled water to the processed soil was 5:1) was prepared, and its conductivity was measured with a conductivity meter (Leici DDS-307A; Shanghai Yoke Instruments Co., Ltd., Shanghai, China). The empirical formula SSC = 0.2882 EC1:5 + 0.0183 was used to calculate the SSC. According to the four grades of saline-alkali soil, the 114 treated soil samples were classified into four types: non-saline soil (0–0.2%), slightly salinized soil (0.2–0.5%), strongly salinized soil (0.5–1.0%), and salinized soil (>1.0%). The sample analysis is shown in Table 1. The SSC of 0–40 cm was the average value of the measured SSC at 0–20 and 20–40 cm, and the SSC of 0–60 cm was the average value of the measured SSC at 0–20, 20–40 and 40–60 cm.

Table 1 SSC of the sampling points.

		Sample size	SSC	
Depths (cm)		G1 (0–0.2%)	G2 (0.2–0.5%)	G3 (0.5–1.0%)	G4 (>1.0%)	Min (%)	Max (%)	M (%)	SD (%)	CV	
0–10	Total	57	42	10	5	0.071	1.460	0.300	0.299	0.995	
MS	38	28	7	3	0.071	1.936	0.293	0.269	0.921	
VS	19	14	3	2	0.074	1.936	0.317	0.354	1.119	
10–20	Total	57	40	11	6	0.074	1.546	0.309	0.303	0.978	
MS	39	26	8	3	0.074	1.484	0.296	0.277	0.934	
VS	19	14	3	3	0.075	1.546	0.332	0.349	1.054	
0–20	Total	62	40	7	5	0.070	2.61	0.297	0.336	1.130	
MS	41	27	5	3	0.070	1.427	0.284	0.271	0.952	
VS	21	13	2	2	0.075	2.61	0.322	0.441	1.369	
20–40	Total	57	43	11	3	0.076	1.448	0.295	0.268	0.908	
MS	38	29	7	1	0.074	1.427	0.302	0.281	0.929	
VS	19	14	4	2	0.076	1.448	0.317	0.304	0.957	
0–40	Total	54	48	10	2	0.070	1.398	0.280	0.228	0.815	
MS	36	32	7	1	0.073	1.196	0.273	0.209	0.766	
VS	18	16	3	1	0.070	1.398	0.300	0.265	0.883	
40–60	Total	54	50	8	2	0.070	1.202	0.278	0.212	0.763	
MS	33	26	5	1	0.070	1.202	0.280	0.221	0.789	
VS	21	24	3	1	0.074	1.099	0.272	0.194	0.713	
0–60	Total	57	46	9	2	0.085	1.386	0.297	0.232	0.781	
MS	37	32	6	1	0.085	1.209	0.292	0.220	0.755	
VS	20	14	3	1	0.076	0.386	0.307	0.257	0.835	
Notes:

MS, modelling set; VS, validation set; G1–G4, non-saline soil, slightly salinized soil, seriously salinized soil, salinized soil; CV, coefficient of variation.

(A) The SSC at the depth of 0–20 cm represents the mean value of the SSC at the depths of S0–10 and S10–20. (B) The SSC at the depth of 0–40 cm represents the mean value of the SSC at the depths of S0–10, S10–20 and S20–40. (C) The SSC at the depth of 0–60 cm represents the mean value of the SSC at the depths of S0–10, S10–20, S20–40 and S40–60.

Remote sensing data acquisition and pretreatment

Sentinel-1, with a short revisit period and a dual-polarization channel (Gao et al., 2020), can provide continuous images of HIA. The radar satellite image used in this experiment was obtained from Sentinel-1 transiting at 13:46 on July 17, 2019, and the imaging time was almost synchronous with the ground sampling time. The mode of Sentinel-1 image used in this experiment was terrain observation by progressive scans (TOPS), in which the corresponding coverage range was 25 km × 25 km, the ground resolution was 5 m × 20 m, the data was of grade one, and the product data was Ground Range Detected (GRD). The dual-polarization channels are Vertical-horizontal (VH) and Vertical-Vertical (VV). Radar satellite image data were downloaded from ESA data access center (https://scihub.copernicus.eu/userguide/).

Sentinel-1 radar remote sensing image was preprocessed in SNAP and ENVI. The image processing process was as follows: (1) precision orbit calibration, (2) thermal noise elimination, (3) radiometric terrain correction, (4) speckle noise removal, (5) geometric terrain correction, and (6) radar backscattering coefficient (VV and VH) extraction. The speckle noise was processed with a filter of Refined Lee and the terrain correction was based on the digital elevation model generated by the Shuttle Radar Topography Mission (SRTM).

Polarization combination index construction and best subset selection

Researches indicated that ground object features can be reflected by the magnitude of radar backscattering coefficient (Yu et al., 2020; Stamenkovic et al., 2017). Due to the limited ground object information in a single backscattering coefficient, in this study, twelve sets of polarization combination indices were generated from the two groups of backscattering coefficients through a series of mathematical operations so as to reflect more ground object information. For convenience, all the polarization combination indices were renamed and shown in Table 2.

Table 2 Nomenclature of polarization combination indexes (PCI).

PCI	New name	PCI	New name	
VH	VH	VV	VV	
VV+VH	V1	VV−VH	V2	
VH/VV	V3	VV2+VH2	V4	
VV2−VH2	V5	VV2+VH	V6	
VV2−VH	V7	VH2+VV2/VV	H1	
(VH2+VV2)/VH	H2	(VV2−VH2)/VH	H3	
(VV2+VH2)/(VV+VH)	H4	(VV2+VH2)/(VV2−VH2)	H5	

Best subset selection (BSS) is used to select variable sets for multiple regression. Its principle is, according to all the different combinations of independent variables, to use partial least squares to perform fit analysis on each combination so as to select the only model whose combination has the best performance. Therefore, BSS is often used in variable selection. Its calculation is as follows: Let K be the number of independent variables (K = 1,2,…, P); 1~P models of predicative variables are fitted. Among the 1~P models, P optimal models were selected when the R2 reached the maximum and RMSE the minimum after adjustment by validation sets. According to the R2 and RMSE after adjustment, an optimal combination of independent variables was selected from P models. The BSS in this study was mainly conducted in LEAPS of R3.5.1.

Construction of salinity inversion models

To ensure the generalization ability and robustness of the model, the total soil samples (n = 114) were randomly divided as calibration set (76 samples) and validation set (38 samples) at the ratio of 2:1. The samples were cross validated using stratified 3-fold cross validation.

Previous studies have shown that the optimal machine learning methods for SSC inversion accuracy vary from research areas (Hoa et al., 2019). Therefore, this study adopted four commonly used machine learning methods: PLSR, QR, SVM and ELM. These methods have demonstrated good universality and stability in soil salinity study (Wang et al., 2020; Hu, Liu & Peng, 2019). Some main characteristics of these methods are outlined as follows.

PLSR models

PLSR is a statistical analysis based on multiple elements (Zhou et al., 2018). As an improved version of least square regression, PLSR enjoys more flexible problem-solving ability because it integrates principal component analysis, multiple linear regression and canonical correlation analysis. Its basic principle is as follows: components are interactively extracted from variables X and Y, and according to principal component analysis, the independent and dependent variable matrices can be decomposed into two matrices multiplied by each other:

(1) Y=U(n×a)Q(a×m)=F

(2) X(n×p)=T(n×a)P(a×p)=E

where n represents the sample size, p is the independent variable, m is the number of dependent variables, a is the number of principal components, E and F are the random error matrix. In model construction and sample prediction, the leave-one-out method was adopted. Cross-validation was used to determine the number of principal components. The establishment and prediction of PLSR model were completed in MatlabR2016b.

QR models

Quantile Regression (QR), proposed by Bassett and Koenker in 1978, is mainly used to study the relationship between conditional quantiles of two variables (Broniatowski et al., 2019). In this study, it handled the relationship between the independent and dependent variables. According to the conditional quantile of the dependent variables, the regression model is derived from fitting. The QR modeling in this study mainly has the following three advantages: (1) It needs no standard normal changes or random perturbations for all variables. It is known from experience that the model accuracy will increase when the variables are perturbed. (2) The abnormal points will not cause too much interference to the model accuracy, so the QR model has a good stability. (3) If the parameters of any quantile are given, the changes of dependent variables caused by the influence of parameters becomes interpretable when the parameters are different. The rationale of QR is as follows: a random sample of the dependent variable Y is {y1,y2,y3,⋅⋅⋅yn}, the sample quantile linearity under the quantile condition τ should meet the following equation:

(3) minβ∈R(∑i=1nyi−xi′β(τ))

where yi represents the dependent variable and β the unknown parameter. For any 0<τ<1, the formula for parameter estimate is:

(4) β′=arg⁡min∑i=1nyi−xi′β(τ)

when the test function is substituted into Eq. (5), which is rewritten as Eq. (6), the parameter x′β(τ) is obtained. Then, the obtained parameter is the unique regression quantile.

(5) minβR[∑i:yi≥xi′β(τ)τ|yi−xi′β(τ)|+∑i:yi≥xi′β(τ)(1)−τ|yi−xi′β(τ)|]

where β(τ) represents the solution of the minimization problem. The QR model establishment and prediction were completed in Eviews 9.

SVM models

SVM, based on statistical theory and structure minimization, employs nonlinear mapping to transform the data into higher-dimensional feature space, in which linear model fit of dependent variables is carried out (Nakagawa et al., 2007). SVM can avoid redundant discrete values of traditional regression models and “over-learning” in variable analysis (Kisi & Cimen, 2012). Recently, the application of SVM has gradually shifted from image recognition and classification to regression problems. The rationale of SVM model is as follows: with the constraint of the condition of

{yi[(ω⋅xi)+b]≥1−ξiξi≥0

(6) (i=1,2,⋅⋅⋅,l)

find the minimum of function

(7) Φ(ω,ξ)=12(ω⋅ω)+c(∑i=1nξi)

where c represents the penalty coefficient used to control the equilibrium of the boundary of error ξ. αi represents the Lagrange multiplier, and it is transformed into a quadratic function, thereby the extreme value is found. With the constraint of

{∑i=1nyiαi=00≤αi≤c

(8) (i=1,2,⋅⋅⋅,l)

find the maximum of

(9) W(α)=∑j=1lαi−12∑i,j−1lαiαjyiyj(xi⋅xj)

Then the optimal discriminant function is

(10) f(x)=sgn{∑i=1lαiαjK(xi⋅xj)+b

where the kernel function K(xi⋅xj) is RBF (the Gaussian radial basis function)

(11) K(xi⋅xj)=exp⁡(−γ‖xj−xi‖2),γ>0

In this study, the SVR analysis of the SVM model was performed in E1071 of R3.5.1, where the kernel function was RBF, Gamma = 0.07, and Cost = 100.

ELM model

As a neural network ELM, proposed by Huang Guangbin (Bai, Huang & Wang, 2016), differs from other neural networks in its feedforward with a single hidden layer (Zhao et al., 2012). Its training process is simple, and the only operation is to set the number of nodes in the feedforward network. Compared with the traditional neural networks, ELM needs no adjustment of the connection weights of the input and hidden layers or the number of valves in the hidden layers. Moreover, this model has strong nonlinear fitting ability and high learning speed. The algorithm is as follows:

(12) φ(x)=h(x)HT(IC+HHT)−1T=[k(x,x1)⋅⋅⋅k(x,xN)]T(1C+ΩEML)−1T

(13) k(a,b)=exp⁡(−‖a−b‖)2/σ

(14) ΩELM(i,j)=k(xi,xj)

where H represents the hidden layer matrix of the neural network, and k is the number of neurons in the hidden layer, which is usually set as RBF nucleus; I is the identity matrix. C is the kernel parameter that needs optimizing; and T is the vector of the predicted target value.

Four machine learning models were constructed with the SSC at different depths as the dependent variables and all the indexes before BSS and the optimal indexes after BSS as independent variables, respectively. The model construction was carried out in R3.5.1.

Evaluation index of model accuracy

The coefficient of determination (R2), root mean square error (RMSE) and correlation coefficient (r) of the model were used for model accuracy evaluation. The formulae of the three evaluation indexes are as follows:

(15) r=∑i=1n⁡(xi−x¯)(yi−y¯)∑i=1n⁡(xi−x¯)2∑i=1n⁡(yi−y¯)2

(16) R2=∑i=1n⁡(y^i−y¯)2∑i=1n⁡(yi−y¯)2

(17) RMSE=∑i=1n⁡(y^i−yi)2n

where xi represents the value of backscattering coefficient, and x¯ the average value of backscattering coefficient; yi the measured SSC value, and y^i the predicted SSC value; y¯ the average SSC; and n the sample number.

In addition, Ratio of Performance to Deviation (RPD) was used to evaluate the model stability. The formula of RPD is as follows:

RPD=∑i=1n(yi−y¯)2n⋅RMSE

Previous studies have divided the models into three categories (Chang et al., 2001): (1) when RPD ≥ 2.00, the model has the highest stability and reliable prediction ability; (2) When 1.40 ≤ RPD ≤ 2.00, the model has good stability and prediction ability; and (3) when RPD < 1.40, the model is unstable.

Results

Division of modeling and validation sets

The collected ground data were divided into four grades in terms of salinity. According to laboratory physical and chemical analysis, all the soil samples were divided as modeling and validation sets. The salt content distribution of the total samples, modeling and validation samples at different depths is shown in Fig. 2.

Figure 2 Sample distribution statistics.

As Fig. 2 shows, the distribution of these three types of samples at different depths is basically the same, indicating the partitioning of modeling and validation sets is reasonable.

Optimal variable combinations by the BSS

According to all the combinations of the given variables, the BSS was used to figure out all the random combinations at the seven depths. Based on these combinations and the R2 of the validation sets, the combinations of the independent variables at the seven depths were identified. The optimal combinations of independent variables (OCIV) at each depth determined by the R2 and RMSE are shown in Table 3.

Table 3 Optimal combinations of independent variables after full subset selection.

SD/cm	NIV	OCIV (P value)	RP2	RMSEp	
0–10	6	VH (0.009), V1 (0.006), V4 (0.007), H1 (0.005), H2 (0.007), H5 (0.03)	0.286	0.364	
10–20	6	VH (0.008), V7 (0.04), H1 (0.005), H2 (0.004), H3 (0.007), H4 (0.02)	0.399	0.363	
0–20	5	VV (0.006), V5 (0.007), V6 (0.04), H4 (0.008), H5 (0.005)	0.678	0.218	
20–40	4	V2 (0.007), V3 (0.005), H2 (0.004), H4 (0.009)	0.737	0.239	
0–40	6	VV (0.007), V3 (0.006), V4 (0.008), H2 (0.006), H4 (0.007), H5 (0.04)	0.846	0.374	
40–60	4	V1 (0.008), V5 (0.008), H2 (0.009), H5 (0.007)	0.674	0.412	
0–60	5	VV (0.005), V5 (0.006), H2 (0.005), H4 (0.007), H5 (0.008)	0.454	0.586	
Note:

SD, depth of soil; NIV, number of independent variables; OCIV, optimal combinations of independent variables.

As Table 3 shows, the two backscattering coefficients plus 12 polarization combination indices, after BSS, generated six optimal independent variables at 0–10, 10–20 and 0–40 cm, respectively. The OCIV were VH, V1, V4, H1, H2, and H5 when the variable combination was at 0–10 cm; the OCIV for 10–20 cm were VH, V7, H1, H2, H3, and H4; and the OCIV for 0–40 cm was VV, V3, V4, H2, H4, and H5. It also shows that as the sampling depth increased the R2 of each depth under the OCIV presented an increase-and-decrease trend and then reached the maximum value of 0.846 at 0–40 cm. As the depth increased, the RMSE first decreased and then increased.

Construction and validation of PLSR-based SCC models

Construction of PLSR-based SCC models

In this section, PLSR is used for model construction. The model accuracy comparison in Fig. 3 shows: (1) the PLSR model based on the OCIV after BSS and the PLSR model before BSS had similar results: both models displayed the best performance when the soil depth was 10–20 cm although their model accuracy was somewhat different; the RMSEc of the modelling set before BSS was 0.09% while the RMSEp of validation set was 0.1%, showing only a slight difference between the RMSEc and RMSEp. (2) Except for 10–20 cm, the model accuracy after BSS was generally lower than that before BSS, which was preliminarily speculated to be related to the decrease of the number of independent variables after BSS.

Figure 3 Comparison of PLSR model accuracy before and after full subset screening.

Validation of PLSR-based SCC models

PLSR models of SSC at different depths were constructed on the basis of the OCIV through BSS (Fig. 4). As Fig. 4 shows, the predicted values were mainly concentrated in zones I and II, and only one predicted value of 0–10 cm was distributed in zone IV. However, no predicted values of 0–10, 0–20 and 40–60 cm appeared in zone III. At 0–10 and 0–40 cm, the R2 of the modeling and validation sets were 0.04 and 0.03, respectively, showing a relatively small difference. At these two depths, the RMSE differences of the modeling and validation sets were 0.12 and 0.03, while the RMSEp and RMSEc were 0.20% and 0.14%, respectively. At 0–20 cm, the difference between Rc2 and Rp2 was the greatest (up to 0.15), and the RMSE difference was also the largest (0.20).

Figure 4 PLSR model based on soil salt content at different depths.

In general, the PLSR model had a satisfying prediction at 0–60 cm. At this depth, the Rc2 of the modeling and the validation sets were 0.47 and 0.35, respectively; the difference between Rc2 and Rp 2 was 0.12 while the difference between RMSEc and RMSEp was 0.02. When the sampling depth was 10–20 cm, the predicted values all appeared in zones I, II and III, and one predicted value was very close to zone IV, and the model fit curve was close to function y = x. At this depth, the corresponding Rc2 and Rp2 were 0.4 and 0.32, the RMSEc and RMSEp were 0.21% and 0.29%, respectively, so there was no overfitting and the model displayed satisfactory prediction. As Figs. 4C–7 shows, at 0–20 cm, the model had the worst prediction performance because the model fit function obviously deviated from function y = x, and the Rc2 and Rp2 were only 0.27 and 0.12, respectively.

Figure 5 Comparison of QR model accuracy before and after full subset screening.

Figure 6 QR models based on soil salt content at different depths.

Figure 7 Comparison of ELM model accuracy before and after full subset screening.

The comparison of all the prediction figures of SSC at different depths showed that, after BSS of OCIV, the fit curve of the measured and predicted SSC at 10–20 cm was closer to the function y = x. The fit accuracy of the PLSR model for 10–20 cm was higher than the predicted values for the other six depths.

Construction and validation of QR-based SSC models

Construction of QR-based SSC models

In this section QR is used for model construction. As the comparison of model accuracy in Fig. 5 shows, the Rc2 and Rp2 both had the best effect before and after BSS at 10–20 cm, and their accuracy differed slightly. At this depth, the RMSEc of the modeling set before and after BSS displayed no clear difference, but the RMSEp after BSS was significantly reduced. At 0–10 cm, both the RMSEc and RMSEp increased, and RMSEp reached the maximum.

Validation of QR-based SSC models

QR was performed on the selected data for different depths to predict the SSC, and the model effect is shown in Fig. 6 below.

As Fig. 6 shows, in the QR models constructed after BSS, most of the predicted values for the seven depths were concentrated in zones I and II. No predicted value appeared in zone IV for 20–40 cm and no predicted value in zone III for 0–60 cm. For other depths, the predicted values appeared in zones I, II, III and IV, which were consistent with the measured ground data, indicating the applicability of QR model.

In the QR models constructed after BSS, the R2 difference of the modeling and validation sets for the seven depths were 0.07, 0.11, 0.12, 0.09, 0.08, 0.07 and 0.06, respectively. The RMSE differences were 0.29%, 0.11%, 0.34%, 0.25%, 0.13%, 0.20% and 0.03%, respectively. The RMSEp of the validation sets (0.62%, 0.51% and 0.53%, respectively) were all over 0.4% at 0–10, 0–20 and 20–40 cm. As is shown, at 20–40 cm, the Rc2 and Rp2 of the QR model were 0.23 and 0.32, and the RMSEp and RMSEc were 0.28% and 0.53%, respectively, indicating poor prediction of the QR model at this depth. At 0–20 cm, the Rc2 of the modeling set reached a maximum of 0.42, the Rp2 of the validation set was 0.35, and there was no overfitting. However, at this depth, the RMSEp and RMSEc of the model were 0.33% and 0.62%, respectively, and the RMSE of the validation set reached the maximum. The RMSE difference between the validation and modeling sets was 0.29%. At 0–20 cm, the R2 of the modeling and validation sets were 0.36 and 0.48, respectively; the maximum difference was 0.34, and the maximum difference of the RMSE was 0.12%, indicating that the QR model for the depth of 0–20 cm was unstable. At 0–60 cm, the Rc2 and Rp2 were 0.40 and 0.46, respectively, and the minimum difference of the R2 was 0.06. At this depth, the RMSEp and RMSEc were 0.25% and 0.28%, respectively, and the minimum difference was 0.03%. At this depth, the model fit curve was close to the function y = x. It showed that the best depth for QR model was 10–20 cm.

Comparison of the QR models for the seven soil depths showed that, after BSS of the OCIV, the predicted SSC values for 10–20 cm appeared in zones I, II, III and IV. Figure 6 also showed that the fit curve of the measured and predicted SSC at 10–20 cm was closer to the function y = x. The results showed that the fit accuracy of the QR model at 10–20 cm was higher than the predicted values of the other six depths.

Construction and verification of ELM-based SSC models

Construction of ELM-based SSC models

In this section, ELM is used to for model construction. As the comparison of model accuracy in Fig. 7 shows, before BSS, the Rp2 was less than 0.2 at both 0–20 cm and 10–20 cm while, after BSS, the Rp2 was less than 0.2 only at 0–20 cm. Meanwhile, the R2 of the modeling set increased over 0.5 at this depth. At 0–60 cm, the Rc2 and Rp2 increased significantly after BSS, but the RMSE changed little before and after BSS. At 40–60 cm, the Rc2 decreased and the Rp2 increased significantly after BSS. However, the RMSE of the ELM model was relatively stable and showed no significant change before and after BSS. As can also be seen, at other depths, the Rc2 and Rp2 were both improved, indicating the suitability of the ELM model constructed after OCIV selection for the SSC estimation at different depths.

Verification of ELM-based SSC models

Figure 8 gives the results of ELM models of SSC at different depths constructed after BSS.

Figure 8 ELM models based on soil salt content at different depths.

After BSS, the ELM models of SSC at the seven depths had no overfitting. The predicted values were concentrated in the vicinity of non-salinized soil and slightly salinized soil in zones I and II. At 20–40 and 40–60 cm, no predicted values appeared in zone IV, indicating the failure of OCIV of the ELM model in predicting the values of salinity after BSS. At 0–10 cm, the model failed to predict the value of severely salinized soil in zone III. However, at other depths, the ELM model had predicted values in zones I, II, III and IV, which preliminarily indicated that the ELM model had better performance than PLSR model in salinity estimation at the seven depths.

Among the models for the seven depths, the model effect was relatively poor at 40–60 cm: the Rc2 and Rp2 were 0.26 and 0.16, respectively, and the RMSEp and RMSEc were both less than the model error of 0.4%. The differences of the Rc2 and Rp2 at 0–10, 0–20 and 40–60 cm reached 0.14, −0.18 and 0.1, respectively, and were relatively large. Meanwhile, the differences of RMSEp and RMSEc reached −0.13%, 0.31% and −0.22%, respectively, and were also relatively large, indicating a poor model stability at these three depths. At 10–20, 20–40 and 0–60 cm, the differences between the Rc2 and Rp2 were 0.09, −0.05 and 0.05, respectively, and were relatively small. The differences of the RMSEp and RMSEc were 0.19%, 0.07% and 0.12%, respectively, and were also small, indicating the stability of the ELM model at these three depths. At 0–40 cm, the difference between Rc2 and Rp2 was 0.04, and the difference between RMSEp and RMSEc is −0.03%, indicating that the ELM model was most stable at 0–40 cm.

In general, at 10–20, 0–20 and 0–60 cm, the fit curve for the ELM models was closer to the distribution trend of function y = x. At 10–20 cm, the Rc2 and Rp2 of ELM model had the best effect, reaching 0.53 and 0.44, respectively. Meanwhile, the RMSEp and RMSEc were 0.29% and 0.10%, respectively, indicating that 10–20 cm was the optimal inversion depth for ELM model.

Construction and validation of SVM-based SSC models

Construction of SVM-based SSC models

In this section, SVM is used to for the model construction. As the comparison of model accuracy in Fig. 9 shows, before BSS, the RMSE of both the modeling and validation sets were relatively high at 0–60 cm while the R2 was relatively low. The R2 of the modeling and validation sets at 10–20 cm was the highest among all the seven depths. Compared with the SVM model before BSS, in the SVM model after BSS, the R2 of the modeling and validation sets at other depths were increased except for the decrease of Rp2 in the validation set at 40–60 cm, and the RMSE of the model decreased and remained stable after BSS.

Figure 9 Comparison of SVM model accuracy before and after full subset screening.

Validation of SVM models

After BSS, the prediction effect of the SVM model for seven different depths is shown in Fig. 10. As is shown, the SVM model based on the OCIV at the seven depths after BSS displayed no overfitting. The predicted values were concentrated in zones I and II, and at each depth, the predicted values appeared in zones III and IV. This trend was also consistent with the distribution of ground measured data, indicating the suitability of the SVM model for SSC estimation at different depths.

Figure 10 SVM models based on soil salt content at different depths.

After BSS, the RMSEs of the SSC inversion models at the seven depths were all less than 0.4%. At 40–60 cm, the SVM model prediction was relatively poor: the Rc2 and Rp2 reached the minimum of 0.37 and 0.23, respectively, and the difference between the RMSEp and RMSEc was −0.19%, indicating the instability of the SVM model at this depth. At 20–40, 0–40 and 0–60 cm, the differences between the Rc2 and Rp2 (0.01, −0.04 and 0.06, respectively) as well as that between the RMSEp and RMSEc (−0.09%, −0.01% and 0.13%) were both relatively small, indicating the relative stability of the SVM model at these three depths. At 0~–10 cm, 10–20 cm, 0–20 cm, the absolute values of the difference of the R2 reached 0.18, 0.11 and 0.13, respectively; and the difference of the RMSE (−0.07%, 0.07% and 0.09%, respectively) were relatively small. At 10–20 cm, the Rc2 and Rp2 reached their maximum of 0.56 and 0.67, respectively; the RMSEp and RMSEc were 0.19% and 0.12%, respectively; and the model fit curve was close to the distribution of function y = x. It showed that the optimal depth for SVM was 10–20 cm.

Model comparison and analysis

With the OCIV at different depths after BSS as independent variables, the inversion models of SSC at different depths were constructed. For the same soil depth, the fit of each model is shown in Fig. 11. Indices of Rc2, RMSEc, Rp2, RMSEp and RPD were used to evaluate the prediction ability of PLSR, ELM, SVM models for different depths so that the optimal inversion model was screened out (Table 4). As is shown, PLSR had the lowest stability (RPD = 1.42) while SVM had the highest stability (RPD = 1.85).

Figure 11 Comparison of measured and predicted SSC based on machine learning.

Table 4 Comparison of soil salinity inversion models at different depths.

Model	Optimal depth (cm)	Rc2	RMSEc	Rp2	RMSEp	RPD	
PLSR	10–20	0.4	0.21	0.22	0.29	1.46	
QR	0–20	0.36	0.17	0.48	0.56	1.63	
ELM	0–40	0.27	0.26	0.23	0.29	1.42	
SVM	10–20	0.56	0.19	0.67	0.12	1.85	

As Fig. 11 shows, the fit curves of PLSR, ELM and SVM models for the depth of 0–10 cm were compact, indicating that the three models have the same estimation ability at this depth. Among the three models constructed for 0–20, 0–40 and 0–60 cm, PLSR had relatively poor fit. The fit curve of SVM was smoother than that of PLSR and ELM at these three depths; and the R2 of SVM model reached 0.57 at 0–40 cm. At 20–40 cm, the fit accuracy of PLSR, ELM and SVM models (with R2 of 0.32, 035 and 0.43 respectively) was relatively high, indicating the stability of the three models at this depth.

The comparison showed that at 10–20 cm the model fit accuracy of PLSR, ELM and SVM reached 0.40, 0.53, and 0.56, respectively, and the three models had the best model fit curve.

Discussion

Estimation performance of the four models

It should be noted that there are few studies on radar remote sensing-based estimation of SSC especially under vegetation due to the lack of a simple and effective radar backscattering model for SSC inversion. Therefore, the use of radar images together with optimal machine learning methods is important for the inversion of SSC under vegetation. For this reason, four machine learning methods were used to model the relationship between Sentinel-1 radar images and the measured soil salinity.

Due to the non-negligible effects of environmental factors such as geographical patterns and agricultural activities on soil salinity, the general relationship between Sentinel-1 signals and soil salinity is highly nonlinear (Bai et al., 2020). This complicated relationship poses a challenge to SSC inversion models. As is known in the previous section, the performance of the four models in validation data set ranked as follows: SVM > QR > ELM > PLSR. The R2 and RMSE of the SVM model reached 0.67% and 0.12%, respectively. Xu et al. (2020) associated pixel brightness of digital images of surface soil with soil salinity, and evaluated the accuracy of PLSR and RF in terms of the RMSE and R2. The results showed that the RMSE of RF was 3.31 g/kg smaller than that of PLSR, the R2 of RF was 0.13 larger than that of PLSR. Chen et al. (2020b) used UAV remote sensing data together with four machine learning methods to build SSC models of sunflower in different growth stages and different soil depths. The comparison showed that SVM, BPNN and ELM outperformed PLSR, and BPNN had the best performance (R2 = 0.718, RMSE = 0.062%).

To sum up, our study showed that nonlinear models (SVM, QR and ELM) outperformed the linear model (PLSR) in prediction accuracy and stability, and this result agrees favorably with that of Taghadosi & Hasanlou (2021) and Wang et al. (2021a). In model construction, the nonlinear models adopted in this study enjoyed the following advantages. The SVM model has good robustness because it can grasp the key samples and eliminate redundant samples. More importantly, it can minimize the sample errors and reduce the upper bound of the model prediction errors using the limited sample information, thus improving its generalization and anti-noise disturbance abilities (Jing et al., 2012). The QR model can more comprehensively characterize the conditional distribution of the explained variables, and the estimation results are more robust to the outliers. The ELM uses the regularization method to calculate unknown variables in order to achieve infinite proximity to the target variables of any continuous system. In general, because of their stronger analytical ability, nonlinear models are more stable and accurate (Farahmand, Sadeghi & Farahmand, 2020). Therefore, they have become powerful tools for SSC monitoring.

Optimal inversion depth

The study showed that 0–20 cm was the best inversion depth of SSC under vegetation. In the study area, the period when the soil was under full vegetation was the middle and later stages of irrigation. In this period, the salt carried by the irrigation water would migrate upwards and aggregate at the shallow soil depths (<30 cm) due to the strong evaporation. The active root layer is in 10–30 cm, so the effect of soil salt on crop growth is closely related to the root zone (Kumar et al., 2021). Therefore, the depth of active root layer is a major reason for the difference of inversion accuracy at different depths. The optimal inversion depth of 0–20 cm is reasonable because this depth is within the active root layer. In the same study area, Tan et al. (2020) also found the optimal inversion depth was 0–20 cm (R2 = 0.875, RMSE = 0.7%) when they monitored soil moisture content in the root zone of mature corn using UAV multi-spectral remote sensing. Because 20–60 cm is beyond the active root layer, the prediction accuracy was unsatisfactory at this depth. Overall, the difference of inversion accuracy in soil depths is not only a statistical result of the inversion models but also the co-result of salt migration and vegetation physiological response.

In this study, the optimal SSC inversion depth of 0–20 cm stemmed from the full consideration of different depths under vegetation as well as the actual irrigation situation in the study area. This result was consistent with that of previous studies. Therefore, the result of this study has more practical value.

Influence of other factors

In this study, soil salinity under vegetation was monitored via Sentinel-1 images of vegetation growth indirectly. However, both soil moisture and salinity have been identified as important factors influencing vegetation growth, especially under water deficit conditions (Wang et al., 2019a; Hassani, Azapagic & Shokri, 2020; Zhang et al., 2021). Therefore, ignoring the effect of soil moisture on vegetation may introduce additional errors in SSC estimation (Liu et al., 2020). Meanwhile, vegetation coverage is also a non-negligible factor affecting the Sentinel-1 signal (Dinh et al., 2018). Existing researches indicate that Sentinel-1 signal is more affected by soil than vegetation in areas with low vegetation coverage (Li, Chen & Xu, 2019). Textural characteristics and roughness of the soil also have some impact on the SSC inversion (Ren et al., 2016; Karthikeyan et al., 2019). In this regard, categorizing the vegetation coverage may be an effective solution to this problem (Lykhovyd, 2021). Therefore, in the future, appropriate models should be selected according to the different categories of vegetation coverage. In addition, human activities can also cause uncertainty in the estimation results (Martínez-Sánchez et al., 2011). For example, the diffuse irrigation used in the study area can exacerbate the extreme distribution of soil salinity, causing great salt accumulation in the topsoil at low topography (Huang et al., 2018; Mao et al., 2020). However, vegetation cover can mask this information, leading to salinity underestimation by the model. In conclusion, future work should focus on coupling existing monitoring models with soil salinity formation mechanisms to achieve more accurate SSC estimation with remote sensing.

Conclusions

In this study, we evaluated the potential of Sentinel-1 and four widely used machine learning methods for estimating SSC at different depths under vegetation in the Shahaoqu Irrigation Area, and reached the following conclusions: The soil salinity in the root zone can be indirectly estimated with Sentinel-1 remote sensing data of the vegetation information.

The best subset selection and different combination indices of backscattering coefficients were helpful to SSC estimation.

The SSC inversion models based on the four machine learning methods achieved satisfactory results. The inversion accuracy of the models ranged as follows: SVM > QR > ELM > PLSR. For SVM, R2 = 0.67, RMSE = 0.12, and RPD = 1.85.

SVM was the best inversion model of SSC at 0–20 and 20–40 cm, and QR was the best at 40–60 cm.

Although the models in the study performed satisfactorily in the inversion of SSC at different depths under vegetation, their accuracy needs further improving. In future studies, introduction of time series or radar data of longer wavelength (S band) may be a better choice for higher accuracy.

Supplemental Information

Supplemental Information 1 Longitude and latitude of the sampling points.

Click here for additional data file.

Supplemental Information 2 The distribution of the total samples, the samples for the modelling sets and validation sets at each depth, respectively.

Click here for additional data file.

Supplemental Information 3 Soil salt content for PLSR models.

The PLSR models are based on the raw data of the measured and estimated values at seven different depths.

Click here for additional data file.

Supplemental Information 4 Soil salt content at different depths for QR models.

The measured and estimated values of the soil salt content at seven different depths for the QR modelling.

Click here for additional data file.

Supplemental Information 5 Soil salt content at different depths for ELM models.

The measured and estimated values of the soil salt content at seven different depths for the ELM modelling.

Click here for additional data file.

Supplemental Information 6 Soil salt content at different depths for SVM models.

The measured and estimated values of the soil salt content at seven different depths for the SVM modelling.

Click here for additional data file.

Supplemental Information 7 Measured and estimated soil salt content.

The measured and estimated values of the soil salt content at seven different depths for the PLSR, QR, ELM and SVM modelling.

Click here for additional data file.

We thank the Key Laboratory of Agricultural Soil and Water Engineering in Arid and Semiarid Areas of Ministry of Education for providing the experiment equipment. We are especially grateful to the reviewers and editors for appraising our manuscript and for offering constructive comments.

Additional Information and Declarations

Competing Interests

Author Contributions

Field Study Permissions

Data Availability

The authors declare that they have no competing interests.

Yinwen Chen conceived and designed the experiments, performed the experiments, analyzed the data, prepared figures and/or tables, authored or reviewed drafts of the paper, and approved the final draft.

Yuyan Du conceived and designed the experiments, performed the experiments, analyzed the data, prepared figures and/or tables, authored or reviewed drafts of the paper, and approved the final draft.

Haoyuan Yin conceived and designed the experiments, performed the experiments, analyzed the data, prepared figures and/or tables, authored or reviewed drafts of the paper, and approved the final draft.

Huiyun Wang conceived and designed the experiments, prepared figures and/or tables, and approved the final draft.

Haiying Chen conceived and designed the experiments, analyzed the data, authored or reviewed drafts of the paper, and approved the final draft.

Xianwen Li performed the experiments, authored or reviewed drafts of the paper, and approved the final draft.

Zhitao Zhang conceived and designed the experiments, performed the experiments, prepared figures and/or tables, authored or reviewed drafts of the paper, and approved the final draft.

Junying Chen analyzed the data, prepared figures and/or tables, authored or reviewed drafts of the paper, and approved the final draft.

The following information was supplied relating to field study approvals (i.e., approving body and any reference numbers):

The field experiments were approved by the Hetao Irrigation District Administration (No.2017YFC0403302).

The following information was supplied regarding data availability:

The raw data for SSC & measured soil salt are available in the Supplemental Files. The construction and comparison of the four models are all based on these raw data of the measured and estimated soil salt content.

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
