# Peer review of "Radar remote sensing-based inversion model of soil salt content at different depths under vegetation"

_PeerJ, doi:10.7717/peerj.13306_

## Round 0.1 · original submission · Major Revisions

The major issues to be addressed in this manuscript are:

1) The discussion section needs to be completely rewritten to consider the results in detail and place them in the context of other research.

2) Provide a justification as to why the microwave data with top soil penetration is able to estimate deeper soil properties.

The authors should also make the various editorial suggestions put forward by the reviewers.

Reviewer 1 ·

Basic reporting

In general, the manuscript "Radar remote sensing-based inversion model of soil salt content at different depths under vegetation” reports on the prediction of soil salt content (SSC) at different depths under vegetation using four different models (PLSR, QR, SVM and ELM) based on radar remote sensing. The manuscript reports on valuable experiment and a significant number of output results.

Experimental design

Experimental design is good.

Validity of the findings

the novelty of presented research is not clear. Please make it clearer. Why you chose all these four models to predict SSC? The comparison of four models is your novelty?
The discussion part is a little bit short. Please check if you could add more details such as a comparison of four models.
A more organized “Conclusions” is needed with a description of your study and the main conclusions, which is also a reflection of your novelty.

Additional comments

Line26-27: “support vector machine regression (SVM)” is supposed to be SVMR for the abbreviation.
Line59-60: "Nurmemet et al. (Nurmemet et al., 2015; Isak & Nurmemet, 2018; Duan et al., 2018)”, did you mean a group of references or just one paper? Please double check.
Line 110: “140mm” and “2000mm” need blank space between number and unit.
Line 133: Please point out the units of distilled water and the processed soil with ratio 5:1.
Line 200: what kind of cross validation did you use for the PLSR?
Line 265-266: the sentence needs to revise “The larger the R2 and the smaller the RMSE, the better the model effect.”
Line 277: after statistical analysis?
Figure 2 needs more annotation.

Reviewer 2 ·

Basic reporting

1. The grammar and logic of this article need to be revised with a native speaker.
2. Please provide a bit more big-picture motivation of how soil salinity analyses threaten society or agriculture or ecology and how they have evolved over the past decade. The paper is missing a "big-picture" introduction with some references in my opinion.
3. The scientific issues in the abstract seem overreaching. Agriculture and soil salinization has been studied in various studies based on remote sensing technologies.
4. The introduction section missing two important elements i.e., (i) Hypothesis explanation, and (ii) Novelty statement. These two important mentionings will guide the reader towards the heart of the article.

Experimental design

1. As mentioned by the authors of the study, the microwave remote sensing data inversion depths are topsoil (0-20), this is because the microwave signal penetration depth is limited, and additionally, the vegetation further reduces the signal. Therefore, it is meaningless to estimate deep soil properties using only the backscattering coefficient and its derived indices.
2. Why have these machine learning algorithms been selected? Even though machine learning models dependent variables using a large number of independent variables, the mechanisms of geographic or remote sensing must exist.
3. Lines 104-115: What are the soil types as per USDA soil taxonomy in this region? And vegetation type?

Validity of the findings

1. Line 174: Why choose the best variables by linear methods and then use machine learning to estimate soil salinity? Two types of methods do not match.
2. Why is the validation set of some models higher than the modeling set? What are the reasons for this phenomenon? What effect does it have on the results?

Additional comments

1. Vegetation affects the backscattering coefficient of soil and is usually removed through optical data. Did the independent variable mention reflect soil or vegetation? It was an indirect estimate? The author should clarify; else, the article's foundation would be lost.
2. In the comparison of models, it is not enough to compare the prediction accuracy, but also to comprehensively evaluate the stability of the model, the degree of dispersion of the results, etc.
3. Lines 48-49: What is the most effective basis? This progression does not sort out well the application of remote sensing technology to soil salinity.
4. The discussions section is terrible. Discussions are not critical writing, do not compare well with prior research, do not explain well the reasons for the results in the paper

---

## Round 0.2 · Minor Revisions

Thank you for taking the time to revise this manuscript.

The reviewers have highlighted a number of points which need to be addressed before it can be accepted. Please make sure these are attended to in detail, as the reviewers have identified that some were not addressed in this revision.

The key issues are:

1) Please provide a review the machine learning literature to justify your choice of methods.

2) Please identify the soils as per the USDA soil taxonomy

3) Expand the discussion to highlight to significance of the study and to compare the model performance with previous studies.

4) Ensure the references are correctly formatted.

Reviewer 1 ·

Basic reporting

I have read the revised manuscript and the authors' responses to my comments. However, there are still some questions need to be answered.
1. For the discussion section, it has been improved comparing with the previous version. However, the comparison with prior research needs to be specific. Most importantly, there is no model performance comparison with other studies, like R2 and RMSE.
For examples, “Compared with linear model (PLSR), nonlinear model (i.e., SVM, QR and ELM) can achieve better performance in prediction accuracy and stability (Taghadosi & Hasanlou, 2021; Wang et al., 2021).” (lines 459-461), it this your study conclusion or the references’ conclusions? Why don’t you give a detail descriptions about the references’ study since you cited?
Again, “This discovery has also been demonstrated by Tan (2020) and Hu (2006).”(line 477-478), what exactly they found? What are the conditions they had, e.g. soil type?
2. The reference format needs to be checked again, different font and italic are found.

Experimental design

None

Validity of the findings

None

Additional comments

1. Line 63: any abbreviation indicated for (HH2 + HV2)/(HH2-HV2)?
2. Line 90: double blank space before “Therefore”.
3. Line 194-195: actually I mean how many fold you used and how many samples you had for each fold.

Reviewer 2 ·

Basic reporting

no comment

Experimental design

no comment

Validity of the findings

no comment

Additional comments

The authors have made changes to the issues raised, and this version is worthy of recognition. However, there are still some problems:
1. Add a literature review of machine learning to the introduction to explain the rationale for choosing these methods.
2. What are the soil types as per USDA soil taxonomy in this region? This question was not answered. USDA soil taxonomy indicates soil order, such as Aridosols, Molisols, and etc.
3. Discussion should highlight the significance of the study.

---

## Round 0.3 · accepted · Accept

Thank you for taking the time to address the queries from the referees. On the basis of your response and the updated manuscript, I am happy to recommend publication.